# Do Poisonous Plants in Pastures Communicate Their Toxicity? Meta-Study and Evaluation of Poisoning Cases in Central Europe

**DOI:** 10.3390/ani13243795

**Published:** 2023-12-08

**Authors:** Sabine Aboling

**Affiliations:** Institute of Animal Nutrition, University of Veterinary Medicine Hannover, Bischofsholer Damm 15, 30173 Hannover, Germany; sabine.aboling@tiho-hannover.de

**Keywords:** co-existence, overgrazing, zero poisoning, co-ingestion, case report, checklist, plant-animal communication

## Abstract

**Simple Summary:**

Secondary plant metabolites, which can be toxins, may exert a repellent function. However, lethal cases of pasture poisoning show that this protection may fail. The reasons behind this are unknown. The aim of this review was (1) to document under which circumstances poisoning occurred in cattle, sheep, goats, and horses and (2) to present a checklist of poisonous pasture plant species that may occur in Central Europe. The checklist comprised 52 taxa, of which 13 taxa were associated with no poisoning owing to the lack of a report, 11 taxa with evidence-based zero poisoning by avoidance or ingestion, and 28 taxa with poisoning due to ingestion. Nine plant taxa caused poisoning in pastures in more than 100 individuals, for example, sycamore (*Acer pseudoplatanus*) in horses, cowbane (*Cicuta virosa*) in cattle, and St. John’s wort (*Hypericum perforatum*) in sheep. Zero poisoning accounted for 40%, and poisoning accounted for 60% of all 85 cases. Poisoning was most often associated with a limited choice of feed (24.7%) and least often associated with the co-ingestion of grass (4.7%). In between were the circumstances of “overgrazing” (12.9%) and “seasonally scarce feed” (10.6%). The results of this inquiry suggest that poisoning virtually always evolves owing to hunger, while 40% zero poisoning proved successful co-existence to a large extent. Therefore, poisonous plants in pastures do communicate their toxicity under animal-friendly grazing management. With the help of the checklist, farmers may evaluate the risk posed by poisonous plants to their animals.

**Abstract:**

One of the possible roles of secondary plant metabolites, including toxins, is facilitating plant–animal communication. Lethal cases of pasture poisoning show that the message is not always successfully conveyed. As the focus of poisoning lies in the clinical aspects, the external circumstances of pasture poisoning are widely unknown. To document poisoning conditions in cattle, sheep, goats, and horses on pastures and to compile a checklist of plants involved in either poisoning or co-existence (zero poisoning), published case reports were evaluated as primary sources. The number of affected animal individuals was estimated within abundance classes from 0 to more than 100. The checklist of poisonous plants comprised 52 taxa. Of these, 13 taxa were deemed safe (no reference was found indicating poisoning), 11 taxa were associated with evidence-based zero poisoning (positive list), and 28 taxa were associated with poisoning (negative list). Nine plant taxa caused poisoning in more than 100 animal individuals. Zero poisoning accounted for 40% and poisoning accounted for 60% of a total of 85 cases. Poisoning was most often associated with a limited choice of feed (24.7%), followed by overgrazing (12.9%), seasonally scarce feed (10.6%), and co-ingestion of grass (4.7%). Hunger interferes with plant–animal co-existence, while zero poisoning improves it. In conclusion, poisonous plants in pastures may communicate their toxicity if the animals have enough alternative feed plants. An individual animal might utterly perceive the communication of toxicity by the plant species but be forced to ignore the message owing to a limited choice of feed options.

## 1. Introduction

Relying on pre- and post-ingestive sensations, herbivorous vertebrates thrive because of their capability to navigate between distinct kinds of plant species within vegetation [1]. This is enabled by the principle that certain volatile and non-volatile molecules act as “secondary plant metabolites, eliciting pharmacological or toxicological effects in humans and animals” [1]. Plant-animal communication is based on these metabolites, on the one hand, and on the senses of the animal, on the other hand. Certain poisonous metabolites that are thought to be species-specific have evolved to permit the co-existence of both certain animals and plants. As the sheer number of studies alone shows, the ecological role of those metabolites as repellents is still almost exclusively demonstrated in the case of non-vertebrates. In herbivorous vertebrates, one assumes a pre-ingestive sensation towards volatile, mostly chemically unidentified metabolites, so-called “attractants or repellents”, leading to a distinct feeding decision [2]. However, in the case of some prominent poisonous plant species, the poison (e.g., oxalic acid in dock (*Rumex*), alkaloids in yew (*Taxus*) and ragwort (*Senecio*) and hypoglycin A in sycamore (*Acer pseudoplatanus*)) neither is volatile nor does it act as a repellent. On the other hand, we know the chemical and functional nature of some potential volatile repellents, such as monoterpenes, which pre-ingestively lead to feed aversions in both naïve red deer calves [3] and lambs [4].

In a post-ingestive sensation, herbivorous vertebrates sense metabolites with their taste organs [5,6]. Among relevant substances that would cause a positive so-called “post-ingestive feedback” [7] are nutrients [8], which act both via taste in the short term [8] and via satiety in the long term [7]. In contrast, some plant secondary metabolites may impose a negative post-ingestive feedback associated either with a bad taste or with illness or pain [7]. For example, drimane sesquiterpenes deter fishes from feeding on sponges and “may be a result of direct action on taste receptors” [9]. Tannins, as non-volatile compounds, are produced by acacias, leading to reduced browsing by giraffes [10]. Pre- and post-ingestive sensations are co-ordinated: deer smell the feed before tasting it [11]. Goats browsing on different species of juniper (*Juniperus*), distinguished by species-specific oil concentrations, did not show whether they distinguished different juniper species through odor, taste, or both [12].

In this respect, poisonous plants’ nutrients may attract a grazing animal, while the plants’ poisons may prevent severe or lethal poisoning by causing adaptive grazing behaviors ranging from complete avoidance to the ingestion of small quantities. The consequences for the animal during adaptation or learning are either invisible as subclinical reactions (no poisoning) or visible as non-specific symptoms, such as diarrhea or salivation (poisoning). Moreover, there are specific symptoms after the ingestion of certain plant species, such as brown urine in the case of pasture myopathy in horses through the ingestion of juvenile *Acer pseudoplatanus*, vomiting in the case of *Rhododendron* poisoning in small ruminants, and colic in horses through the ingestion of *Colchicum autumnale*.

Unfortunately, lethal cases of pasture poisoning show that protection by secondary plant metabolites may fail. Corresponding symptoms and diseases in the animal are described in detail, while other variables in the context are scarcely described, such as the biology of poisonous plants as well as the composition and state of the vegetation. This approach has the following three consequences:(1)Plants are regarded toxicologically. Publications on poisonous plants rely on experimentally proven toxicities mostly in laboratory animals. This approach is termed “intoxication” and describes the artificial process of the targeted and forceful application by humans of a toxin to an animal. Moreover, feeding experiments serve to define the risk based on a dose-effect relationship. This leads to entries, such as “[w]ild plants that can cause poisoning in farm animals” [13], or to conclusions, such as “[b]ased on toxicological data, poisoning of livestock and companion animals by plants is a relatively common occurrence in Europe” [14]. Even though a study acknowledges the “[r]esponse of herbivores” to plant poisons, it lists records of “[t]oxic plants in Central Europe” [15].(2)Case details are neglected. In a recent (2015) publication, poisoning with autumn crocus (*Colchicum autumnale*) is described as follows: “[p]oisoning may occur when the young spring leaves or autumn flowers are ingested in pastures […] (3)” [16]. The indicated reference (3) above was published in 2012 and states “[p]oisoning of animals in the spring involves ingestion of the young leaves, whereas in the autumn the flowers of plants growing wild in pastures are implicated (Humphreys, 1988)” [17]. Humphreys [18] finally indicates the primary sources [19] and [20], stating that “*C. autumnale* poisoning in cattle is dealt with by Debarnot [20] and in sheep by Tribunskii [19]”, but does not mention any details. These references from 1968 and 1970 revealed that the “animals” were lambs and cattle and that one single cow ate “flower after flower” on her pasture [20]. None of the references mentioned “young spring leaves”.(3)Context of poisoning is not recorded. A review from 2010 on “animal poisoning in Europe” listed “[…] one case report of sheep poisoned […] by monks hood (*Aconitum napellus*)” [21]. The indicated source stated “[u]nfortunately not specific information could be collected at that time” [22]. Already in 2008, both those “notorious discrepancies of thinly documented information” were addressed, and the monography “International Poisonous Plants Checklist: An Evidence-Based Reference” was published along with original literature references, covering mainly cases of conserved feed [23]. In 2015, the first approach appeared, entitled “Exposure assessment of cattle via roughages to plants producing compounds of concern”, that explicitly considered not only toxins as a variable in pasture poisoning but also ecologically relevant aspects such as the abundance of plant species in grasslands and the selection behavior of animals within trough-fed roughage [24].

These issues (toxicological focus, neglected case details, missing context) make it difficult to be aware of the outer circumstances under which the species-specific relationship between plants and herbivorous animals fails such as the kind and quality of feed plant species available on the pasture, the composition of the vegetation, or the presence of certain developmental stages of poisonous and feed plants. Therefore, animal keepers would emphasize rather incalculable dangers than the mutual benefits of grazing for both husbandry animals [25] and agricultural landscapes [26]. To evaluate the risks for animals on pastures, evidence-based data from everyday practice need to be compiled. Among the references, scientific case reports enjoy priority since they enable an inductive approach with authentic first-hand data. The aim of this review is (1) to document under which circumstances poisoning occurred in cattle, sheep, goats, and horses and (2) to present a checklist of poisonous pasture plant species in Central Europe from the point of view of these animals.

## 2. Materials and Methods

The pastures considered in this study were mainly located in Central Europe, geographically including, according to “Flora of Central Europe” [27], the countries Austria, Croatia, the Czech Republic, Germany, Hungary, Luxemburg, Poland, Slovakia, and Switzerland. Moreover, this study evaluated a few cases observed in the United Kingdom, in Norway and Sweden, and in Spain and Italy. Here, “pasture” is understood as any kind of grassland such as continuously growing herbaceous vegetation within a fenced area that is regularly grazed by large herbivores, no matter whether or not the site is mown. Pure meadows and year-round wild pastures are excluded due to their deviating floristic composition.

Scientific databases of references were browsed for plants, using the terms “poisonous” and “toxic” as well as “cattle”, “sheep”, “goats”, and “horses” as search algorithms. Recorded plant taxa for the checklist needed to meet three criteria: (1) presence of secondary plant metabolites such as toxins that are potentially poisonous for farm animals, (2) growing in pastures, and (3) occurring in Central Europe. In the case of chemically similar species of one genus or family, only one or two genera appeared in the checklist. In the case of chemically similar and closely related genera, both genera were listed as one entry.

Apart from primary sources such as case reports, papers, or brief communications on pasture poisoning, ecological or agricultural studies on grazing farm animals contributed to the checklist, too. In two of three cases, personal communications were included and labeled as such. In order to reproduce the authentic information from the references as authentically as possible, the results consisted largely of citations.

“Poisoning” comprised all surviving or lethal cases of pastured animals with reported clinical symptoms of poisoning due to a known ingestion of poisonous plants. While the names of the plant secondary metabolites are listed (Appendix A), the corresponding symptoms of poisoning are not systematically described but are often part of the citations. Two kinds of references were regarded as proof of poisoning: (1) small formats, such as a correspondence as documented evidence, and (2) peer-reviewed papers as rationale, which provided evidence. However, it was not always possible to deduce whether certain sources had been peer-reviewed or not, particularly those published before 1950.

“Zero poisoning” (the absence of poisoning) comprised all cases of pastured animals with direct contact with poisonous plants but without known (reported or documented) symptoms of plant poisoning. Zero poisoning was either indirectly assumed due to the lack of a case report or directly proven on the basis of any publication as defined below (Table 1). If proven via a publication, zero poisoning was based either on the documented avoidance or on the documented inconsequential ingestion of a poisonous plant by an animal. If the authors did not reasonably report whether the animal had avoided or ingested the plant, an unknown reason for zero poisoning was assumed. In any case, published evidence-based incidents of zero poisoning were scrutinized to see whether the authors had indeed failed to mention any clinical or subclinical symptoms.

The number of affected animal individuals in the checklist was estimated within abundance classes from 0 up to more than 100, cumulatively adding records documented in as many papers as could be found.

An example for criterion B “0 cases proven” is the following: “[s]uch animals [goats] are usually kept on marginal land where bracken [*Pteridium aquilinium*] can occur. As there seem to be no reports about the effect of bracken on goats and the extent, if any, to which bracken carcinogens may be transferred to the milk, such studies would be very valuable” [28]. The corresponding combination “goat and *Pteridium*” was recorded in this meta-study with both the reference [28] and the circumstance “unknown”.

If both “zero poisoning” and “poisoning” had been detected in a certain plant-animal combination because of two distinct reports in two papers, the checklist would categorize the corresponding plant species as “poisonous” due to priority. However, the finding of “zero poisoning” with those plant species under different circumstances was supplemented (see Section 3.2.8).

Pigs as dominating important husbandry animals were omitted because only four studies evaluating plant-pig interactions on pastures were available ([29,30,31,32]).

Each identified plant species was searched for a relation to cattle, sheep, goats, and horses in order to check whether the combination led to a result. Grazing always had to be the dominant kind of feeding, but data on pasture animals stabled only during the winter served likewise as material. Cases of pasture poisoning counted as proven if the authors provided a substantial report, not merely anecdotal notes. Thus, statements such as “[p]oisoning due to eating roots of water hemlock is discussed” [30] were dismissed since it was unclear whether the cases they referred to had been pasture poisoning or not. Moreover, several field observations were revealed to not be cases of pasture poisoning and thus were not considered. For example, “P. described the illness of a bullock which died within three min. of eating water hemlock roots [*Cicuta virosa*] contained in mud from a recently cleaned out drinking place […]” [31].

In particular, papers that provided inconclusive evidence were excluded. For example, a case report on hound’s tongue (*Cynoglossum officinale*) reads (translated from German) as follows: “In a group of four horses, one horse showed severe flatulence along with symptoms of colic after some days of being pastured. Hound’s tongue was not identifiable at the time of the site’s inspection [three weeks later after the incidence] neither as rosettes nor as flowering shoots” [33]. Instead, the plant species had been identified only in the feed sample provided to the authors before the inspection of the site took place. Due to the contradictory findings, it remains open whether hound’s tongue caused the symptoms or not.

Taxonomy of botanical scientific names followed “The International Plant Names Index and World Checklist of Vascular Plants 2023” [34]. The genus *Senecio* has been recently partly renamed in *Jacobaea* [34]. However, in the case of *Senecio alpinus* (now *Jacobaea alpinus* (L.) Moench.) and *Senecio jacobaea* (now *Jacobaea vulgaris* Gaertn.), these synonyms were still used here due to their widespread usage. The dictionary of plant names by Zander (2002) [35] provided the common names.

## 3. Results

The checklist of poisonous plants comprised 52 taxa (see Appendix A). Related to four animal species, respectively, this resulted in 208 combinations (52 × 4), denoted as “cases” (Table 2).

A total of 169 of these combinations (81.3%) represented assumed or proven zero poisoning (A–B). However, for most of the combinations (64.9%), no publication could be found. These cases counted as assumed zero poisoning (A). In contrast, a further 34 cases of zero poisoning were documented by at least one publication and counted as proven zero poisoning (B). Moreover, 39 cases of plant poisoning existed (C–E). Among the references that gave evidence in all 73 cases (B–E) were three personal communications from the author [36,37,38].

### 3.1. Checklist of Poisonous Plant Species in Pastures in Central Europe

Among the 52 taxa of the checklist, 13 taxa were associated with assumed zero poisoning: field maple (*Acer campestre*), mugwort *(Artemisia vulgaris)*, Barbara’s herb (*Barbarea*), hoary alison *(Berteroa incana*), adderwort *(Bistorta officinalis*), cuckoo flower (*Cardamine pratensis*), common horsetail (*Equisetum arvense*), spurge (*Euphorbia*), ground ivy (*Glechoma hederacea*), bristle grass (*Setaria*), comfrey (*Symphytum*), clover (*Trifolium*), and vetch (*Vicia*).

The remaining 39 plant taxa were involved in the above-mentioned total number of 73 evidence-based cases of both zero poisoning (11 taxa) and poisoning (28 taxa). In detail, the following taxa caused no poisoning: marsh marigold (*Caltha palustris*), giant hogweed (*Heracleum mantegazzianum*), horseshoe vetch (*Hippocrepis comosa)*, poppy (*Papaver*), lousewort (*Pedicularis palustre*), smartweed/knotgrass *(Persicaria/Polygonum*), yellow rattle (*Rhinanthus*), tansy (*Tanacetum vulgare*), meadow rue (*Thalictrum*), globeflower (*Trollius europaeus*), and white veratrum (*Veratrum album*). Known ingestion of the following 28 plant taxa led to poisoning in at least one of the four animal species: sycamore (*Acer pseudoplatanus*), common box elder *(Acer negundo*), monkshood (*Aconitum napellus*), caraway (*Carum carvi*), fat hen (*Chenopodium album*), cowbane (*Cicuta virosa*), autumn crocus (*Colchicum autumnale*), hound’s tongue (*Cynoglossum officinale*), bugloss *(Echium vulgare*), horsetail, marsh horsetail *(Equisetum palustre*), sweet grass (*Glyceria)*, hogweed *(Heracleum sphondylium*), curled-leaved St. John’s wort (*Hypericum triquetrifolium)*, St. John’s wort (*Hypericum perforatum*), cat’s ears (*Hypochaeris radicata)*, bog asphodel (*Narthecium ossifragum*), hemlock water-dropwort (*Oenanthe crocata*), parsnip (*Pastinaca sativa*), canary grass (*Phalaris*), bracken (*Pteridium aquilinium*), buttercup (*Ranunculus*), dock (*Rumex*), alpine ragwort (*Senecio alpinus*), tansy ragwort (*Senecio jacobaea*), common groundsel (*Senecio vulgaris*), black nightshade (*Solanum nigrum*), yellow oat grass (*Trisetum flavescens*), and nettle (*Urtica dioica*).

Taking into account the number of animal individuals concerned, ingestion of nine plant taxa correlated with poisoning on pastures in more than 100 individuals: sycamore (*Acer pseudoplatanus*): horses; *Cicuta virosa* (cowbane): cattle; autumn crocus (*Colchicum autumnale*): cattle and sheep; St. John’s wort (*Hypericum perforatum*): sheep; bog asphodel (*Narthecium ossifragum*); bracken (*Pteridium aquilinium*): cattle and sheep; dock (*Rumex*): sheep; tansy ragwort (*Senecio jacobaea*): horses; and yellow oat grass (*Trisetum flavescens*): cattle.

### 3.2. Circumstances of Plant Poisoning on Pastures

In total, seven scenarios, Nos. 1–7 (categories, circumstances), were distinguished based on evidence-based (proven) cases:No ingestion of poisonous plants—no poisoning (variant “avoided” of zero poisoning);Ingestion of poisonous plants—yet no poisoning (variant “accepted” of zero poisoning);Poisoning associated with seasonally scarce feed;Poisoning associated with limited choice of feed;Poisoning associated with overgrazing;Poisoning associated with co-ingestion of grass;Poisoning or zero poisoning under unknown circumstances.

Some plant species that caused poisoning under the circumstances numbered 3–7 (see Section 3.2.1–3.2.7) were involved in proven zero poisoning, too, and were presented as a supplementary category (see Section 3.2.1,Section 3.2.2,Section 3.2.3,Section 3.2.4,Section 3.2.5,Section 3.2.6 and Section 3.2.7).

When the variable “circumstance” was added to the plant-animal combination, it turned out that there were several instances of the same plant-animal combination happening under more than one circumstance (for example, tansy ragwort (*Senecio jacobaea*) and horses, associated either with overgrazing (scenario No. 5) or co-ingestion (scenario No. 6)). Therefore, the number of cases increased from 73 to 85. Zero poisoning accounted for 40% (n = 34 cases) and poisoning accounted for 60% (n = 51 cases) (Figure 1).

Poisoning was most often associated with limited choice of feed plant species (24.7%) and happened the least with co-ingestion of grass (4.7%). Surprisingly, zero poisoning along with acceptance of a poisonous plant species turned out to be the second most common circumstance (17.6%). In contrast, animals avoided poisonous plants less frequently (10.6%). Both “overgrazing” (12.9%) and “seasonally scarce feed” (10.6%) only belonged to a medium-frequency class of circumstances. Representative examples of the 85 cases are presented in the following (Section 3.2.1,Section 3.2.2,Section 3.2.3,Section 3.2.4,Section 3.2.5,Section 3.2.6 and Section 3.2.7).

#### 3.2.1. No Ingestion of Poisonous Plants—No Poisoning (Variant “Avoided” of Zero Poisoning)

Avoidance (e.g., ostensibly successful plant-animal communication) was observed for seven plant taxa in a total of nine cases (Table 3).
Cattle“*Caltha* [*palustris*] is not eaten at all by cows” [39].*Papaver*: “[a]nimals are safe since the unpleasant odour and taste of the plants render them obnoxious (Long 1910)” [42].Sheep*Equisetum palustre*: “sheep were strongly selective, that means, shoots from marsh horsetail were intentionally left” [41] (translated).Goats
*Veratrum album* was observed to be avoided by goats [26].Although goats “nibbled” creeping buttercup (*Ranunculus repens*), lesser celandine (*R. ficaria*), and bachelor’s buttons (*R. aconitifolius*) [46], they avoided lesser spearwort (*R. flammula*) [26].Horses*Colchicum autumnale*: “[m]ost interviewees (76.2%) with horses reported that their horses neither feed on *C. autumnale* on the pasture […]” [40].*Rhinanthus*: “[y]et there seems to be a lack of clinical data regarding actual poisonings. That might be because it’s not consumed at all (have heard 3rd hand that a horse will spit it out), not enough is consumed, or perhaps mechanically little is picked up at haying” [45].

#### 3.2.2. Ingestion of Poisonous Plant—Yet no Poisoning (Variant “Accepted” of Zero Poisoning)

No evidence of health problems or poisoning was reported in 15 cases in which the animals ingested 12 poisonous plant species (Table 4). Ingestion of these species was confirmed by direct observation of grazing behavior according to the sources cited below.
Cattle*Acer pseudoplatanus*: “Since the seedlings were not selected but always eaten along grasses and forbs, we observed their ingestion as a byproduct of grazing. […] there is not only a direct proof of consumption on the field level by various distinct field methods but also on the chemical level as metabolites of HGA [hypoglycin A] and MCPrG (methylenecyclopropylglycine) are further detected in urine and milk samples […]” [47].*Persicaria/Polygonum*: “[c]attle, sheep, and goats eat *Polygonum bistorta* willingly, while horses avoid it (Wyżycki 1845)” [26].Sheep and goats*Ranunculus*: “[e]specially sheep and goats like to nibble at poisonous plants such as Ranunculus aconitifolius, Ranunculus ficaria, Ranunculus repens, Rhinanthus spp., Rumex acetosa, Rumex alpinus […]” [46].*Heracleum mantegazzianum*: “[b]y contrast grazing is an environmentally safe control method. Sheep grazing sustains a dense, short vegetation of forbs and grasses (Andersen, 1994). […] For a complete eradication of Giant Hogweed a high grazing pressure is needed. Grazing by 10 sheep per ha was found to change the vegetation towards a less species rich community dominated by grazing tolerant species […]” [49].*Tanacetum vulgare*: “[w]hat! Sheep control tansy (Tanacetum)?” [52].*Senecio jacobaea*: “[s]heep continuously preferred ragwort. The daily intake was above the currently assumed lethal dose, varying between 0.2–4.9 kg per sheep. Clinical, hematologic, and blood biochemistry parameters mostly remained within the reference limits. Initially elevated liver copper content declined over time” [51].Horses*Equisetum palustre*: “feeding traces showed no selectivity towards marsh horsetail” [41] (translated). This study was published in 2012. In contrast, another source from 1952 reported that “poisoning with the green plant is hardly to be observed since horses avoid the weed in pastures” [53] (translated).

#### 3.2.3. Poisoning Associated with Seasonally Scarce Feed

Seasonally scarce feed meant either too little feed during times of drought or inadequate feed outside the core pasture season. In any case, pasture biomass was extremely reduced, causing nutrient deficiency in farm animals over long periods. Moreover, the concentration of plant secondary metabolites was higher in certain developmental stages in two plant species involved in that scenario: (1) In springtime, the concentration of hypoglycin A in *Acer pseudoplatanus* was higher in seeds and seedlings than in the adult leaves [54]. (2) In summertime, the concentration of alkaloids in *Senecio jacobaea* was higher in flowers than in the leaves [55]. Under such circumstances, nine cases of poisoning happened on pastures with nine plant species involved (Table 5).
Cattle*Chenopodium album*: “[a]t present [1975] the west country is experiencing drought conditions. Unbelievably for us [in Great Britain], grass is in very short supply […]” [59].*Pteridium aquilinium*: “[…] bracken [poisoning […] in the year 1893 […], when a drought, ‘the severest of the century’ (Penberthy, 1893) [in Great Britain], lasted through the spring and summer and dried up everything except the bracken to which the cattle turned as the only food available […]. The disease is commonest in dry years when other herbage is scarce […]” [63].*Narthecium ossifragum:* “[a] combination of factors, including the scarcity of grass caused by previous dry weather, may have contributed to this poisoning incident in County Fermanagh in the summer of 1989. […] In the present outbreak of poisoning, the drier areas of the farm were grazed preferentially by the cattle […]. The dry summer and aeration of the peat bog due to mechanical turf cutting operations may have contributed to the proliferation of bog asphodel on the site” [61].*Oenanthe crocca*: “[d]rought and poor grazing may also compel the cattle to seek more nutritious grass in marshy places in many of which *Oenanthe crocata* grows in abundance […] The roots […] are attractive to cattle because of their parsnip-like appearance and also because of their taste and odor […] during a period of very dry windy weather which greatly reduced the nutritional value of the grass. Five Ayrshire heifers were found dead in a wooded, boggy area to which they had gained access by breaking through the boundary fence. The only significant finding at post-mortem examination was the presence of pieces of undigested *Oenanthe crocata* root in the rumen of each animal […] [a] considerable number of roots were lying free on the shingle and all that was left of some was the growing stem about one inch high. There was no grass growing between the plants and the sea and it was concluded that the turbulence in the water during the storm had washed away the sand and gravel from the roots which were left exposed by the receding tide” [62].*Equisetum palustre*: “[...] they had found that the general condition of the cattle may deteriorate during the cool and rainy grazing periods, when they become more susceptible to *E. palustre* poisonings. In these cases there are usually other factors contributing to the sickness” [60].Sheep*Aconitum napellus*: A flock of 20 healthy animals was released for the first time of the year in May, when the grass had not yet grown exceptionally. Instead, the sheep fed on *Aconitum napellus* that grew within the neighboring garden, accessible to the sheep. The plants were just a few centimeters high. All seven sheep with symptoms fully recovered [58]. “The plant is not usually eaten (acrid test), and field poisoning is uncommon” [6].“*P[halaris] tuberosa* can cause death in cattle, but this occurs much more frequently in sheep on fresh growth of the plant after rain, especially at the end of a dry season in cool weather when this plant shoots more rapidly than other species […]. Hungry sheep are most likely to be poisoned […]” [18].Horses*Acer pseudoplatanus:* “[b]ecause sprouts are more common in Spring and seeds are more common in Autumn, it would appear that horses eat more seeds than sprouts either because more seeds are available or because they prefer seeds to sprouts. […]. In general, Spring pasture contains more and better grass than Autumn pasture, and thus horses may have less reason to eat other feedstuff” [57].*Senecio jacobaea*: “The occurrence of the case under discussion was preceded by a period of drought, and there had been but little grass on the pastures for some weeks” [65]. And another source reported: “There was one conspicuous feature common to the majority of affected farms, and particularly striking in several instances, and that was, that the grazing was of a very poor quality, the very short, good grasses and clovers being very scarce, and that plants commonly regarded as weeds were common. In general, in years with low rainfall, cases of ragwort poisoning in grazing horses increased […] there were also two further peaks of incidence: first in March, when the working horses feed on hay instead of straw because of the start of the working season on the fields. Second in August when the work is over and the horses spend their free time on poor pastures” [64].

#### 3.2.4. Poisoning Associated with Limited Choice of Feed

In contrast to circumstances of seasonally sparse grass, limited choice meant that the biomass of 1 of the following 17 plant species dominated in 21 cases (Table 6).
Cattle*Trisetum flavescens*: This species made up the largest portion of biomass, between 18 and 25%, in pastures with a high diversity (10 grass and 21 herbaceous species) [82]. Except for *Anthriscus* (5–18%), *Dactylis* (8–11%), and *Trifolium* (3–18%), all other species reached biomasses between 1 and 8% [82]. In contrast, this grass species was missing on extensively exploited pastures without any fertilization and animal movements as well as on intensively utilized pastures with abundant fertilization and frequent movement of cattle [81]. “On pastures on the alp, the symptoms gradually diminish, however, appear more intensive in the following winter. There is little yellow oat grass on the alp pastures, yet much in the dales” [83] (translated).*Cicuta virosa*: “[…] [on] low-lying, marshy land where *C. virosa* grows abundantly. The authors stated that cattle find the plant palatable and are attracted by its smell. “[…]. Cattle should be prevented from grazing in the neighbourhood of water reservoirs on low-lying pastures especially during drought” [67].*Colchicum autumnale*: “the vegetation consisted mainly of rushes [Juncus] and coverage of *Colchicum* reached 40% on 50–100 m^2^” [68] (translated).*Echium vulgare*: “[t]en 1-year-old fighting bulls died between October and March from a herd of 700 animals […]. All animals had grazed in the pastureland […]. Close inspection of the pastureland where animals were grazing revealed large quantities of *Echium vulgare* (80%) and *Senecio vulgaris* (15%) […]” [70].*Narthecium ossifragum*: “[d]uring the summer of 1992, an apparently new disease with renal failure as its dominating sign occurred in grazing cattle in western Norway (Fl¢yen et al., 1993) The animals were grazing uncultivated pastures and *Narthecium ossifragum* was a common plant in all the areas where the disease occurred. Two hundred and thirty-two animals suffered from the disease and 137 died” [84].*Senecio jacobaea*: “[s]everal inspections of these pastures in June and July 1997 revealed that *S*[*enecio*] *alpinus* was very common” [78]. “Ten out of 75 calves died or had to be slaughtered after they had grazed on wasteland carrying a heavy growth of *S. jacobaea*” [80].*Cynoglossum officinale*: “[t]owards the end of April, 1959, a herd of 23 Friesian cows in various stages of pregnancy gained access to an area of waste land and had been grazing there for several hours before their escape was discovered. […] It was noticed immediately that the cattle were grazing an area heavily contaminated with hound’s tongue or dog’s tongue in the leafy stage of growth […]. Heavy cropping of the plants had taken place. […] The rumen contained large quantities of the leaves of the hound’s tongue […]” [69].*Hypericum crispum* “is very eagerly eaten by herbivores in Turkey; therefore, one cannot stop husbandry animals from ingesting this plant in large quantities during each summer on fields, ditches, and dried meadows as the only grazing sites” [72] (translated).*Ranunculus*: “A cow on a pasture with a heavy crop of buttercups (*Ranunculus bulbosus*) became ill, with salivation, coughing, restlessness and rigors. It recovered the next day” [75]. “After having been housed and given good feed, cows in late pregnancy were turned out onto fields where the pasture was poor and contained buttercups (*Ranunculus sp*.). Within 14 days clinical signs including tympany, diarrhoea with blood and mucus, rapid pulse and difficult breathing developed and 5 animals died or were killed shortly before or after calving, despite symptomatic treatment. Consistent postmortem findings were intestinal haemorrhages, fatty liver and fat infiltration of kidneys” [76]. “[…] two heifers, apparently developed a taste for *R. sceleratus*. Having just recovered from acute poisoning, they returned to the same place and began to eat the buttercups again; they had to be removed from the field for their own safety” [18].Sheep*Pteridium aquilinum*: “[s]heep are sometimes so reluctant that they seem ready to starve rather than eat the bracken. […] [T]he sheep had little choice but heather or bracken to eat. Thirteen sheep had been lost in a few weeks in a flock of only 43 ewes” [63]. “Since 1950 cases of bracken poisoning with sheep increased, while the frequency of burning down heather decreased, leading to a fiber-rich herbage that the sheep avoided in favor for bracken as only available feed.” [63] There were “[…] only heather and a few other plants such as bilberry while the slopes are heavily covered with bracken concealing a poor carpet of grass. The observation that is experimentally it is very difficult to persuade sheep to eat bracken fits in the general impression that sheep avoid it” [63].*Chenopodium album*: three out “[o]f 40 ewes transferred after shearing to a sown pasture in which the grass had failed and *C. album* predominated, 1 died and 2 had to be slaughtered” [66].*Narthecium ossifragum*: “[p]hotosensitization occurs annually in the Blackface sheep on three hill farms in Perthshire. […] Shepherds are insistent that only lambs become affected. […] An examination of the pastures was carried out by Dr. D. Martin of the West of Scotland College of Agriculture. He reported that the pasture was a typical hill sward. *Molinia* was dominant and *Narthecium ossifragum* was present. When the wetter parts of the hill were examined closely, *N. ossifragum*, the bog asphodel, was seen to be more common than originally thought. The spike-like leaves protruded through tufts of *Sphagnum* and many of the tips had been bitten off” [74].*Hypericum perforatum*: “in a flock with 700 ewes, kept within a large paddock, almost 200 animals were fallen sick within three days. On this extensive grazing system, a large portion of *Hypericum perforatum* was present. While being tented, St. John’s wort is being avoided or eaten in small amounts only. In contrast, keeping sheep on paddocks showed that the animals had ingested St. John’s wort down to the stems” [73] (translated).*Rumex acetosella*: “in a flock with more than 600 ewes, a chronic disease was observed, marked by oedematous heads and throats as well as by increasing emaciation. It has been observed that *Rumex acetosella* was present to a great extent on the pastures and made up more than 50% of the vegetation on some spots” [73] (translated).
Horses*Heracleum sphondylium*: “[t]he animal [horse] had been moved 10 days previously to a pasture that had not been grazed by animals for a number of years. On inspection, around 50 per cent of the pasture consisted of hogweed plants […]” [71].*Senecio alpinus*: “during inspection of the associated mountain pasture, apart from a species-poor vegetation of grasses, a dominating contamination with *S. alpinus* on a large scale was found within a partly tall-grown vegetation on a strongly eroded, endangered, northern exposed site. The moist soil had been fertilized through intensive grazing activities. Alpine ragwort, densely growing in groups, had been increasingly eaten. In contrast, single plants remained often untouched” [79] (translated).

#### 3.2.5. Poisoning Associated with Overgrazing

“Overgrazing” meant that feed sources were exhausted because pastured animals had ingested the whole biomass of acceptable plants. Under such circumstances, they had started to feed on 10 poisonous plant species in 11 cases (Table 7).
Cattle*Aconitum napellus*: “One single case series shows a repeated, although exceptional, individual behavior towards monkshood (*Aconitum napellus*). Each year from 1995 to 1999, exactly between 9 July and 15 July, 1 heifer died out of a herd of 80 pastured animals. On the 20th of July, each year, the herds were brought to another pasture. “Toxicosis usually occurs during a ‘toxic window’ or when little else is available to eat” [86].*Glyceria aquatica*: “one cow, found dead, unfortunately had ingested a fair amount of sweet grass (*Glyceria aquatica*) that was present at the riverbank since there was a shortage of grass that has become rare on the meadow” [88] (translated). “Starting point was the intake of reed sweet grass by the [pregnant] heifers, ingested out of particular need for energy. Adapted to wet sites, this grass species possesses broad, fleshy leaves and forms lush vegetation that reaches a height of some 200 cm […]. Also due to its low fibre (3.89%) and high protein content (6.46%) with a dry matter content as low as 23.2% it represents an attractive food before flowering” [87].*Pteridium aquilinum*: “There is no doubt that bovine bracken poisoning causes serious losses to hill farmers in some years —when the ecological conditions are conducive to its appearance. I have seen a Welsh farmer lose 22 pedigree Friesian heifers—his whole herd replacements valued at £2000 in 1950, with it. […] he had been asked by his own son when passing a badly bracken infested hill, ‘why don’t those cattle get bracken poisoning?’ And he had replied, ‘John, I have never had bracken poison there and cattle have grazed on that hill all my years here.’ That night a message had been received about the same cattle saying there was suspected bracken poisoning. Sixty-six heifers were grazing on that hill and 1 had died. The remainder were immediately moved on to another field and every animal was treated with batylalcohol as a preventative: some animals which became ill were treated again with batyl alcohol as a curative but with absolutely no effect. The result was that out of 66 animals grazing on that hill 54 died and some died from bracken poisoning more than a month after being removed from the hill.” [90]. In another case, dietetic reasons were suggested: “[…] an exceptional outbreak in which some 30 cattle died when several hundred were placed on a new[-seeded] pasture in sustained wet weather […]. The cattle here left the young lush grass and deliberately sought the young bracken shoots as if in search of some change, possibly a more fibrous food” [44].*Senecio jacobaea*: “I generally observe five to six cases every year in animals [cattle] commonly from about 12 to 15 months old. It is usual for one case only to occur on any farm in this district, but occasionally two cases of the disease may occur, and in very occasional outbreaks several animals may be affected. […] In this instance the animal [cattle] was allowed to graze for a few hours each day in pasture which was very short of grass, it being in the very early spring. Most of my observations are confined to the occurrence of the disease in cattle on pasture during the spring and summer seasons. The grazing is always short and land frequently overstocked” [64].Goats*Urtica dioica*: “in midsummer on an almost bare ground, a flock of goats was kept on some 5000 m^2^. One individual showed swollen parts of the mandible. Parasitosis was excluded since feces control revealed no infestation. After the ground had been checked botanically, the owner reported that this goat regularly consumed the nettles as the only plant species still available in large amount. After the goats had been given access to a new pasture, the symptoms abated” (translated; personal communication) [37].Horses*Ranunculus*: “[o]n a pasture consisting mainly of the buttercups Ranunculus acris and Ranunculus sceleratus, a 4-year-old horse developed paresis of the hind quarters” [77].*Senecio jacobaea*: “I have abundant evidence that over-stocking and under-feeding have the greatest possible influence in the production of the disease [with draught horses]” [64]. “I have observed the disease in two animals, one a 5-months-old foal and the other an aged gelding, which had been grazed on three fields of temporary grass in September, 1928, the foal having died a few weeks before its older companion. The dam of the foal and a number of young cattle had also been grazed on these fields, which were small in acreage with grass very short and full of weeds, the most conspicuous being the first and second year old stages of the common ragwort plant […]. The dam is still alive, and has so far presented no symptoms indicative of liver cirrhosis” [64].*Acer negundo*: “the meadow [in mid-April] was very bare and I had recommended to put the ponies calmly onto the bare meadow since they were well fed” (personal communication) [38] (translated).*Hypochaeris radicata*: “[f]ood scarcity along with a relatively increased coverage of *H. radicata* (up to 20%) could explain the appearance of the disease in the late season on the grazing areas investigated” [89].*Pastinaca sativa*: “on a 2-hecatare pasture, having been grazed for years by some 10 horses without problems, one horse with a white-skinned mouth developed severe photodermatitis in August. At that time, the grasses had been eaten down to a lawn-like vegetation of 3 cm in parts, from which plentiful fruiting shoots of parsnip protruded. On a representative area of 100 m^2^, 548 apical parts of shoots were missing, corresponding to 91% of the whole number of shoots counted. Since the remaining basal parts of the shoots showed significant signs of damage through grazing, many horses had probably ingested the fruiting parsnips; however, the only white-skinned animal had been suffering from photodermatitis the first time” (translated; personal communication) [36].

#### 3.2.6. Poisoning Associated with Co-Ingestion of Grass

Poisoning associated with co-ingestion of grass occurred in four cases when grasses covered four poisonous plant species that animals incidentally co-ingested (Table 8).
Cattle*Equisetum palustre*: “[i]nevitably, the heifers ingested not only reed sweet grass but marshhorse tail, too, which was growing in between with a relative biomass of 5%. Thus, they had no choice to avoid a plant that cattle normally do otherwise. The haemorhagic enteritis that the young cows developed, i.e., the symptoms of palustrin poisoning, is congruent to symptoms in a feeding trial: Cattle reacted to 34.7 g TS marsh horsetail/100 kg BM/d with diarrohea [5]” [87].*Senecio alpinus*: “moreover, cows are not able to distinguish the still young plants from freshly growing grass in spring or after the first hay cut, and do ingest them (Duby, 1975)” [91] (translated).Horses*Acer pseudoplatanus*: “[…] the authors documented purely the spontaneous ingestion that happened rather unintentionally with stage 2 seedlings, in particular when the horse seemed to be attracted by lush grass wherein these seedlings were more or less hidden. The latter was also true for stage 4 seedlings. Although during the whole observation time of nearly two months (8 April to 31 May), there was an increasing presence of stage 4 seedlings […] and a corresponding higher and denser grown grass that might have distracted the horses from stage 4 seedlings, the young woody plants were almost invisible in the thick lush vegetation typical in spring, yet still as short as the grass layer” [54].*Senecio jacobaea*: “[i]n the outbreak described here, the clinical and laboratory findings in exposed horses are presented together with epidemiological evidence that pasture rather than hay was the principal source of the alkaloids. […] Examination of the pasture in the spring of 1982 revealed a patchy, but in places very heavy, growth of young ragwort plants at the same height as the sward. […] Groundsel (see *vulgaris*) was also present in moderate amounts. […] Little ragwort was observed on these pastures in previous seasons and the finding of many small ragwort plants emerging with the grass in the spring suggests a recent heavy establishment of the plant in this pasture, horses perhaps grazing it inadvertently with the grass […]” [92].

In all cases of co-ingestion, animals were in particular need of energy either due to the season (spring and fall) or due to pregnancy (Figure 2).

#### 3.2.7. Poisoning or Zero Poisoning under Unknown Circumstances

In 16 cases, information was missing as to the circumstances of whether the plant had been eaten or not (cases of zero poisoning) or as to the conditions of the pasture when the plant had been consumed (cases of poisoning). Thirteen plant species were involved in ten cases of zero poisoning (with nine plant species) and six cases of poisoning (with four plant species) (Table 9).
Cattle*Cicuta virosa*: “[a] neighbouring area had been dressed with ammonia fertilizer which could have disturbed the sense of smell of the cattle. Possibly the cattle had mistaken the plant for a root crop” [95].*Colchicum autumnale*: “11 cases of poisoning: (1) June: in trenches many *Colchicum* plants, one dead animal; (2) May: Half of the pasture covered with *Colchicum*, one dead, four sick; (3) month not indicated: six dead animals on a pasture with grazed *Colchicum* plants; (4) June: four dead animals on a pasture with grazed *Colchicum* plants; (5) month and pasture conditions not indicated: one sick animal; (6) month not indicated: four dead animals with *Colchicum* in the rumen; (7) May: three sick bulls on a pasture where some *Colchicum* plants were grazed; (8) month and pasture conditions not indicated: four sick cows; (9): month and pasture conditions not indicated: one sick cow; (10) September: one cow, new in the region, ate *Colchicum* flowers [in the original “veilleuses de nuit”]; (11) Spring: three dead animals on a pasture with abundant *Colchicum* plants” [20] (translated).*Narthecium ossifragum*: “The disease occurred on pasture which had been limed at the end of July 1989 and immediately stocked with 25 suckler cows and their calves, aged two to three months. By mid-August, 15 of the cows were in poor body condition but continued to graze. Over the following two weeks these cows became anorexic and rapidly lost body condition; 11 became recumbent and either died or were euthanased on welfare grounds, but the other four recovered” [61].Sheep*Colchicum autumnale*: “[a]lmost all the animals in a large nomadic herd of sheep developed clinical signs of intestinal irritation and intermittent diarrhoea with mucus and sometimes blood, as they crossed a swampy region densely populated with meadow saffron (*C. autumnale*). The sheep were exhausted and hungry and 2 that died were depressed and had difficult, noisy breathing” [97]. “A high annual incidence of *Colchicum autumnale* (meadow saffron) poisoning usually occurs in May to July in lambs grazed on the highlands of the Central Tien-Shan in Kirghizia” [19].*Trisetum flavescens*: “[a] [Kentucky blue grass] *Poa pratensis*, [rough meadow grass] *P. trivialis*, [rye grass] *Lolium perenne*, [meadow fescue] *Festuca pratensis*, *Trisetum flavescens*, [meadow oat grass]; *Helictochloa pratensis, Avenula pratensis*, [cock’s foot] *Dactylis glomerata* and [white clover] *Trifolium repens* sward at 680–720 m alt. was rotationally grazed by a mixture of heifers and sheep (2 livestock units/ha) or by heifers and sheep alternately. Alternate grazing produced greater av. daily liveweight gains in cattle (an additional 105 g/d) whereas mixed grazing improved sheep liveweight gains (an additional 8 g/d). DM herbage yields under both grazing systems were similar at 8 t/ha” [105].Goats*Solanum nigrum*: “[o]n examining the pasture, considerable quantities of a weed were found, and it was obvious that it had been freshly eaten” [103].Horses*Senecio jacobaea*: “on the basis of a pathological-histological finding, a poisoning with common ragwort has been assumed. Larger quantities of this plant were found on the pasture, and the hay was also obtained from the affected meadows” [100] (translated). “The animals concerned were all riding horses with an average age of 15 years. The pastures were of bad quality with a large amount of common ragwort. There was concern that an estimated four horses had ingested common ragwort on the pasture” [101] (translated).*Cicuta virosa*: “[p]ortions of cowbane (*Cicuta virosa*) stems were found in their stomachs, and the bases of chewed-off plants were found in the field” [96].

#### 3.2.8. Cases of Proven Zero Poisoning with Plant Species that Have Caused Poisoning in Other Cases

Cases of proven zero poisoning of the whole animal population with plants that had yet to cause poisoning in other cases were thus not counted as zero poisoning but as poisoning. Therefore, the following five cases did not appear in the above-presented figures and tables.
*Equisetum palustre* and cattle: “Thus, the animals first seek selectively seek out the areas less covered with marsh horsetail to feed. Marsh horsetails were often found on or in the avoided tufted sedge. However, it was not possible to determine whether the animals had selectively eaten the marsh horsetail in particular” [41] (translated).*Senecio jacobaea* and cattle: “A herd of some 15 cattle grazed on pastures where ragwort was dominating and a control group of another 15 animals on a pasture without ragwort. After one year, no individuals showed any elevated liver enzymes or clinical symptoms” (translated) [107].*Senecio alpinus* and cattle: “unquestionably, the animals grazing on the pasture do not take this plant voluntarily, as long as there is enough feed. It could be clearly observed during the inspection of the farm in the canton Schwyz that tufts of alpine ragwort stood up tall above the other plants in the meadow and were neither partly nor completely ingested” [91] (translated).*Hypericum* and sheep: as already cited (Section 3.2.4): “while being tended, St. John’s wort is being avoided or eaten in small amounts only” [73] (translated).*Senecio jacobaea* and horses: “An example of yearlong grazing of horses on pastures with ragwort revealed that the animals, both young and adult ones, did have co-ingested the plant in small amounts, but did not fall ill” (translated) [108].

## 4. Discussion

The aim of this review is (1) to document and evaluate the circumstances of pasture poisoning in the most common husbandry species, cattle, sheep, goats, and horses, and (2) to present a corresponding checklist of poisonous pasture plant species in Central Europe.

### 4.1. Hunger as Driving Force in Cases of Pasture Poisoning

Apart from only six cases of poisoning under unknown circumstances (7.1%; see Figure 1) as well as one case of assumed dietetic reasons (cattle and *Pteridium*; see point 3 in Section 3.2.5), the data revealed that severe health problems or fatal poisoning due to poisonous plant species evolved practically always due to hunger as a natural, indeed banal [109], consequence of ingesting everything available regardless of the quality. This finding was argued as early as 1926: “The conditions under which animals are tempted to eat poisonous plants are of considerable importance. We have still much to learn on this subject, and the greater part of the necessary information can be collected and recorded only by workers in the field. Excessive hunger is doubtless the main cause of the consumption of poisonous plants, and that arises from a relative scarcity of other foodstuffs, which is due most commonly to particular weather conditions, drought especially. Placing too many animals on a given grazing area will obviously have the same effect of inducing a shortage of other and more palatable foodstuffs. Animals which have been travelling are particularly liable to eat poisonous plants; probably this is often merely a matter of hunger. […] the first step should be to determine whether the plant will cause harm when ingested by animals, and whether any animal is likely ever to consume an effective quantity under any combination of natural conditions” [110]. Most often the starting point is insufficient feed choice, while the number of cases of overgrazing seems surprisingly low against such a background. However, the single kinds of circumstances are hardly to be distinguished: seasonally scarce feed would result in overgrazing which leads in turn to insufficient feed choice or co-ingestion of poisonous plants. Examples such as the following one reveal that pasture poisoning is a multivariate phenomenon, too: “[...] they had found that the general condition of the cattle may deteriorate during the cool and rainy grazing periods, when they become more susceptible to [marsh horsetail] *E*[*quisetum*] *palustre* poisonings. In these cases there are usually other factors contributing to the sickness” [60].

Co-ingestion is an exceptional phenomenon in various regards. It is not only an extreme symptom of hunger, but also always associated with a special physiological need for energy of the animals involved. In contrast to other scenarios, not scarce but lush grass turns out to be the problem in all variants of this kind of poisoning, except in *Acer* seeds that most often fall on sparse grass in the fall. While ingesting the lush grass, animals obviously accept or do not fully sense repellents in ragwort (*Senecio*), marsh horsetail (*Equisetum palustre*), or juvenile seedlings of sycamore (*Acer pseudoplatanus*) hidden in the nutritious feed, although at least horses can distinguish between gradients of repellents. Horses ingested 19.1% of juvenile seedlings with 1.65 mg total phenolics/g fresh weight (FW), yet only 5.46% of older seedlings with 8.48 mg total phenolics/g FW [54]. Hunger would lower any tolerance towards unpalatable plants in order to survive, in particular in distressed animals: “[Blister buttercup] *R*[*anunculus*] *sceleratus* is most likely to give rise to problems, owing to its occurrence in damp places where it can provide lush vegetation when the rest of the pasture is comparatively bare [or covered with unpalatable sedges]” [18]. Even for monkshood (*Aconitum*), the species with one of the most potent toxins in plants in Europe [111], the tolerance level may be lowered; however, the tolerance level differs individually, as the corresponding case reports in sheep (see Section 3.2.5) and cattle prove (see Section 3.2.3). Only a few animals of the whole flock were affected (7 of 20 sheep, all recovered; 1 of 80 heifers died). These data confirm that the threshold for the tolerance of poisonous plants is fundamentally individual, just as the tolerance of hunger or the threshold of perception of repellents and other plant signals is individually pronounced [112]. Hunger, as described above, obligatorily interferes with any plant-animal communication and changes animal behavior, resulting in a dysfunctional acceptance of unpalatable plants in certain animal individuals or in entire herds. Zero poisoning with plants such as horse marshtail (*Equisetum palustre*), tansy ragwort (*Senecio jacobaea*), and alpine ragwort (*Senecio alpinus*) in cattle or St. John’s wort (*Hypericum perforatum*) in sheep, causing poisoning under other circumstances, shows that poisonous plants are not automatically but potentially poisonous, depending not upon the toxins but on the circumstances.

According to the data of publications, poisoning of entire cattle herds with cowbane (*Cicuta*) in 1955, autumn crocus (*Colchicum*) in 1968 and 1975, bog asphodel (*Narthecium*) in 1988, bracken (*Pteridium*) in 1964 and 1965, and yellow oat grass (*Trisetum*) in 1980 is a historical finding in Central Europe. Pastured cattle are becoming scarcer [113], while the number of horses on paddocks has risen steadily during the last 50 years [114]. As an interesting fact, poisoning of large horse populations has occurred two times: in cart horses in the 1930s with tansy ragwort (*Senecio jacobaea*) and in the present millennium in pleasure horses with sycamore (*Acer pseudoplatanus*).

### 4.2. Do Poisonous Plants in Pastures Communicate Their Toxicity?

The proportion of 40% zero poisoning in all evidence-based cases is surprisingly high in a database where cases of zero poisoning are underrepresented due to their lack of clinical relevance. This figure shows that plants and animals on pastures can co-exist to a large extent. Obviously, plants with plant secondary metabolites have successfully mediated their message of being harmful or not to grazing animals even within the confined conditions of fenced pastures.

Zero poisoning happens both via avoidance and acceptance and is based on pre- and post-ingestive mechanisms [1], mentioned in the introduction. However, as late as 1952, one could explain zero poisoning in horses of pastures with *Equisetum* only with avoidance. Physiological (safe) ingestion of poisonous plants is enabled by both behavioral and physiological mechanisms. First, the kind and amount of ingested poisonous plants depends upon species-appropriate socialization [115] and sustainable experiences [116] of young animals. Second, the species-specific tolerance relies on the detoxification mechanism. For example, cattle feed on *Acer* seedlings due to their tolerance of hypoglycin A [117], and sheep tolerate pyrrolizidine alkaloids of their preferred feed plant *Senecio jacobaea* [118]. In goats grazing *Rumex*, a mechanism of adaptation to oxalate digestion seems to be responsible [119]. However, in the case of horses grazing marsh horsetail (*Equisetum palustre*), we know neither the tolerance nor the detoxification level, nor the pre-ingestive pattern of sensation, and even not whether the responsible toxin or repellent is palustrine, thiaminase, silicate, or another compound [120] and whether it acts as a kind of “feeding brake”. The same is true for most of the plant species, including those with potent toxins, such as giant hogweed (*Heracleum mantegazzianum*), tansy (*Tanacetum vulgare*) (both eaten by sheep), and white veratrum (*Veratrum album*) (avoided by goats). The variety of plant secondary metabolites is astonishing and their functions in particular for vertebrate herbivores are widely unknown: Did the 120 phenolic compounds in older sycamore seedlings on horse paddocks [54] contribute more effectively to the higher rate of avoidance by horses than the 82 compounds in young seedlings? Is there any synergistic effect? Are there certain compounds more relevant than others? At what concentration is a single compound perceived?

In contrast, assumed (not evidence-based) cases of zero poisoning probably cannot be exclusively traced back to successful plant-animal communication for two reasons: First, some combinations are unlikely to happen: goats would avoid all plants on wet sites such as marsh marigold (*Caltha*), cowbane *(Cicuta*), and sweet grass (*Glyceria*) because they simply avoid the water-flooded sites of those plants. Second, zero poisoning probably includes overlooked cases of asymptomatic or subclinical poisoning, too: “[…] it is difficult to assess the full effects of plant poisonings, as the toxic compounds may cause only indistinct signs such as mild digestive disturbances or reduced fertility, that pass more or less unrecognized” [1]. Moreover, the tolerance level towards hunger or repellents might be highly variable and individual, as summarized as early as 1947 “[o]ur knowledge of the whims and reactions of the grazing animal—his psychology in short—is limited and elementary in the extreme […]” [121].

On the basis of the circumstances analyzed here, as long as there is feed choice, even scarce feed is not a problem: “[i]t is remarkable that I have never observed a case of the [liver] disease [through tansy ragwort (*Senecio jacobaea*) in horses] on a mountain farm, where tillage is naturally very limited, where the bulk of the grazing consists of permanent virgin pasture, and where the hay commonly used is of the meadow variety” [64]. An unspectacular co-existence is typical for numerous popular plant species with plant secondary metabolites such as sour dock (*Rumex acetosa*). The plant contains oxalic acid that may bind calcium in the blood, a process that can lead to hypocalcemia [122]. Although sour dock (*Rumex acetosa*) grows on almost every pasture in Central Europe, no cases of poisoning are known. Another prominent example is ground ivy (*Glechoma hederacea*), which is poisonous to horses in the case of forced feeding [123] but obviously never eaten on pastures since no report exists (assumed zero poisoning) [109].

### 4.3. Checklist of Poisonous Plants as a Tool for Risk Management

The checklist comprises 52 species and might not be complete when one considers the species richness of European grasslands. Its advantage is both the incorporation of suspicious plant species that were revealed to be no problem for grazing animals and the exclusively empirical basis due to documented or mainly peer-reviewed case reports. In other words: The checklist contains cases of symptomatic poisoning (negative list) as well as cases of zero poisoning (positive list). Moreover, the checklist includes cases of assumed zero poisoning under the premise that no reference—neither of poisoning nor of zero poisoning—is available. The inclusion of assumed and proven zero poisoning along with proven poisoning is the only possibility for completely documenting the state of the art on an empirical basis of evidence-based cases. This may be of practical relevance, too, since funding programs of the European Union would not only encourage the grazing of husbandry animals but also promote floristic diversity on grassland. In this context, farmers encounter the dilemma that with diversity, the number of poisonous plants increases as well [124]. However, with the availability of data on the circumstances of cases, farmers can avoid accidental pasture poisoning in the case of certain plants (negative checklist) and may evaluate the risks of certain other plants (positive checklist). There might also be potential benefits of poisonous plants (weeds) on pastures since “[m]any poisonous plants in small doses are medicinal, such as [red knees] *Persicaria hydropiper*, [columbine meadow rue] *Thalictrum aquilegiifolium* and others” [125]. One can even speculate that these unpalatable plants such as poisonous species and weeds might stimulate movement in pleasure horses since the the animals will need to walk further in search of better feed—as long as there is something better.

## 5. Conclusions

“The viper, though it kills with it, does not deserve to be blamed for the poison it carries, as it is a gift of nature” [126]. From the point of view of the plant, poisonous plant species may become dysfunctionally toxic under two circumstances: if non-voluntarily ingested on pastures with scarce feed plants or if unintentionally co-ingested by animals in special need of energy.

These findings are all the more relevant as floristic diversity became a goal of modern grassland management during the past decade [113]. In particular, the possibility of free feed choice for husbandry animals has been recently emphasized, too [127]. Floristic diversity along with both provision of adequate feed and avoiding overgrazing would not only offer feed choice between nutritive, dietetic, and poisonous plants but also facilitate successful plant-animal co-existence on pastures. This meta-study of the circumstances of pasture poisoning reveals that plant-animal communication inevitably fails most often simply due to feed deficiency. An individual animal might utterly perceive the communication of toxicity by the plant species but be forced to ignore the message due to a limited choice of feed options. This is different from a failure in communication.

Although there is still an insufficient understanding of the role of secondary plant metabolites for herbivorous vertebrates, the question of whether poisonous plants on pastures communicate their toxicity in general can empirically be answered in the affirmative.

## Figures and Tables

**Figure 1 animals-13-03795-f001:**
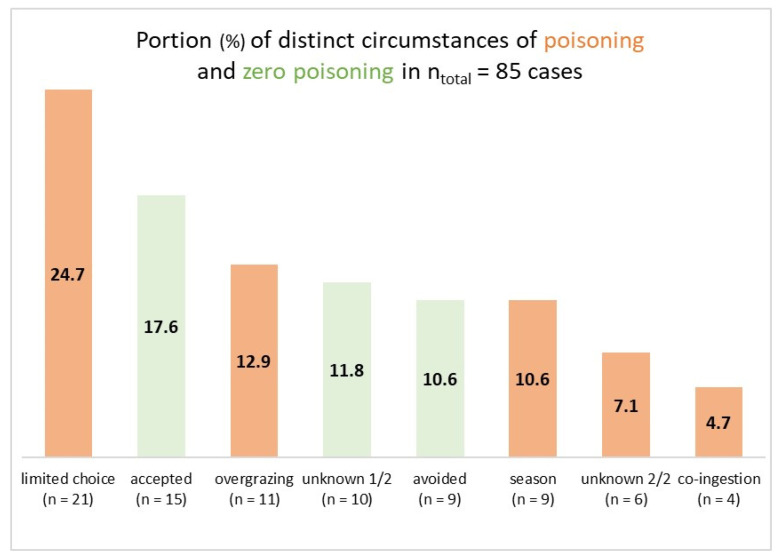
Distribution of poisoning (red columns) and zero poisoning (green columns) in percentage and number of cases. Only proven cases were considered (B-E; see Table 2). Abbreviations of the circumstances: “limited choice” = poisoning associated with limited choice of feed; “accepted” = no poisoning, yet ingestion of poisonous plants; “overgrazing” = feed sources were exhausted; “unknown 1/2” = circumstances of zero poisoning were not described; “avoided” = no poisoning, no ingestion of poisonous plants; “season” = poisoning associated with seasonally scarce feed; “unknown 2/2” = circumstances of poisoning were not described; “co-ingestion” = grass covered the poisonous plant that animals incidentally co-ingested.

**Figure 2 animals-13-03795-f002:**
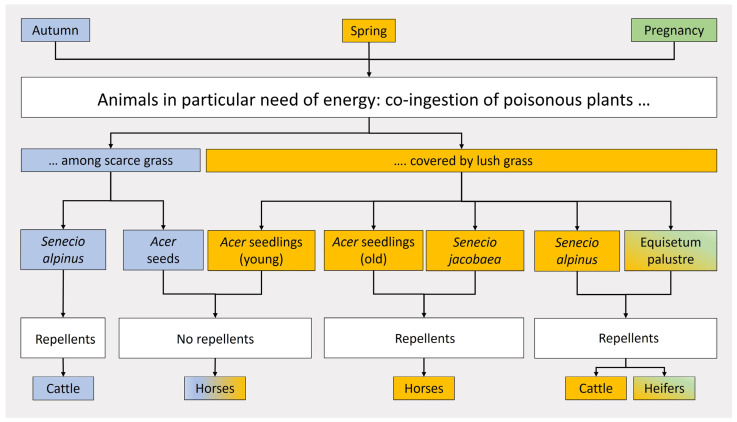
Co-ingestion as a circumstance in distinct plant and animal species. Repellents are assumed on the basis of observations of grazing husbandry animals in the case of older *Acer* seedlings (phenols), *Senecio* (sesquiterpenes), and *Equisetum* (silicate). For references of the cases, see Table 8.

**Table 1 animals-13-03795-t001:** Criteria for data selection and corresponding number of individuals.

Label	Criteria of Data Selection and Definitions of Poisoning and Zero Poisoning	Animal Individuals (n)
A	0 cases assumed or yet to be documented if no publication reference could be found that mentioned the presence of both the plant species and the animal species on a pasture (indirect evidence): “assumed zero poisoning”. In other words, from the lack of such a publication reference, the lack of a poisoning incidence has been assumed.	0
B	0 cases proven or yet to be documented (“proven zero poisoning” or synonymous “evidence-based zero poisoning”) if a publication reference on the presence of both plant and animal species on a pasture was detected (direct evidence), with the proof of (1) avoidance of the plant, (2) ingestion of the plant, or (3) of any other kind of co-existence leaving unanswered whether the plant was avoided or ingested.	0
C	1–9 individuals in proven cases of pasture poisoning, mentioned in one or more publications.	<10
D	10–99 individuals in proven cases of pasture poisoning, mentioned in one or more publications.	>10
E	100-more individuals in proven cases of pasture poisoning, mentioned in one or more publications.	>100

Labels A–E are added for a better orientation.

**Table 2 animals-13-03795-t002:** Animal individuals and cases of plant poisoning per animal species.

Label	Animal Individuals (n)	Cases (n)	Cases (%)	Cattle Cases (%)	Sheep Cases (%)	Goat Cases (%)	Horse Cases (%)
A	0 (assumed)	135	64.9	50.0	65.4	80.8	63.5
B	0 (proven)	34	16.3	15.4	17.31	15.4	17.3
C	<10	20	9.6	15.4	7.7	3.8	11.5
D	>10	8	3.8	9.6	0.0	0.0	5.8
E	>100	11	5.3	9.6	9.6	0.0	1.9
A–E	Total	208	100	100	100	100	100
A–B	0 (assumed + proven)	169	81.3	65.4	82.7	96.2	80.8
B–E	0 > 100 (proven)	73	35.1	50.0	34.6	19.2	36.5
C–E	1 > 100 (proven)	39	18.8	34.6	17.3	3.8	19.2

The letters A and B denote two kinds of evidence. The letters C-E denote distinct numbers of animal individuals. The labels are used to show at a glance the content of the three formed groups A–B, B–E, and C–E.

**Table 3 animals-13-03795-t003:** Zero poisoning by avoidance of poisonous plants, indicated by reference numbers.

No.	Plant Species, Common Name	Plant Species, Scientific Name	Cattle	Sheep	Goats	Horses
1	Marsh marigold	*Caltha palustris*	[39]			
2	Autumn crocus	*Colchicum autumnale*				[40]
3	Marsh horsetail	*Equisetum palustre*		[41]		
4	Poppy	*Papaver*	[42]			[42]
5	Buttercup	*Ranunculus* ^(1)^			[26]	
6	Yellow rattle	*Rhinanthus*		[43,44]		[45]
7	White veratrum	*Veratrum album*			[26]	
		Cases (n)	9			

^(^^1)^ related to lesser spearwort (*Ranunculus flammula*).

**Table 4 animals-13-03795-t004:** Zero poisoning when the plant species was eaten, indicated by reference numbers.

No.	Plant Species, Common Name	Plant Species, Scientific Name	Cattle	Sheep	Goats	Horses
1	Sycamore	*Acer pseudoplatanus* (juv.)	[47]			
2	Marsh horsetail	*Equisetum palustre*				[41]
3	Giant hogweed	*Heracleum mantegazzianum*		[48,49]		
4	Lousewort	*Pedicularis palustre*			[26]	
5	Smartweed/knotgrass	*Persicaria/Polygonum*	[26]	[26]	[26]	
6	Buttercup	*Ranunculus* ^(1)^		[46]		
7	Yellow rattle	*Rhinanthus*			[46]	
8	Dock	*Rumex*			[46]	
9	Tansy ragwort	*Senecio jacobaea*		[50,51]		
10	Tansy	*Tanacetum vulgare*		[52]		
11	Meadow rue	*Thalictrum*	[26]			[26]
12	Globeflower	*Trollius europaeus*			[26]	
		Cases (n)	15			

^(1)^ related to creeping buttercup (*Ranunculus repens*), bachelor’s buttons (*R. aconitifolius*), and lesser celandine (*R. ficaria*).

**Table 5 animals-13-03795-t005:** Cases of poisoning associated with seasonally scarce feed, indicated by reference numbers.

No.	Plant Species, Common Name	Plant Species, Scientific Name	Cattle	Sheep	Goats	Horses
1	Sycamore	*Acer pseudoplatanus* (juv. and seeds)				[56,57]
2	Monkshood	*Aconitum napellus*		[58]		
3	Fat hen	*Chenopodium album*	[59]			
4	Marsh horsetail	*Equisetum palustre*	[60]			
5	Bog asphodel	*Narthecium ossifragum*	[61]			
6	Hemlock water-dropwort	*Oenanthe crocata*	[62]			
7	Canary grass	*Phalaris*		[18]		
8	Bracken	*Pteridium aquilinium*	[63]			
9	Tansy ragwort	*Senecio jacobaea*				[64,65]
		Cases (n)	9			

Number of affected animal individuals (n): beige: n < 10; orange: n > 100.

**Table 6 animals-13-03795-t006:** Incidents of poisoning associated with limited choice of feed plant species, indicated by reference numbers.

No.	Plant Species, Common Name	Plant Species	Cattle	Sheep	Goats	Horses
1	Fat hen	*Chenopodium album*	[66]	[66]		
2	Cowbane	*Cicuta virosa*	[67]			
3	Autumn crocus	*Colchicum autumnale*	[68]			
4	Hound’s tongue	*Cynoglossum officinale*	[69]			
5	Bugloss	*Echium vulgare*	[70]			
6	Marsh horsetail	*Equisetum palustre*				
7	Giant hogweed	*Heracleum sphondylium*				[71]
8	Curled-leaved St. John’s wort	*Hypericum triquetrifolium*	[72]	[72]		[72]
9	St. John’s wort	*Hypericum perforatum*		[73]		
10	Bog asphodel	*Narthecium ossifragum*		[74]		
11	Bracken	*Pteridium aquilinium*		[63]		
12	Buttercup	*Ranunculus*	[75,76]			[77]
13	Dock	*Rumex*		[73]		
14	Alpine ragwort	*Senecio alpinus*	[78]			[79]
15	Tansy ragwort	*Senecio jacobaea*	[80]			
16	Common groundsel	*Senecio vulgaris*	[70]			
17	Yellow oat grass	*Trisetum flavescens*	[81]			
		Cases (n)	21			

Number of affected animal individuals (n): beige: n < 10; light red: n > 10; orange: n > 100.

**Table 7 animals-13-03795-t007:** Incidents of poisoning associated with overgrazing, indicated by reference numbers.

No.	Plant Species, Common Name	Plant Species, Scientific Name	Cattle	Sheep	Goats	Horses
1	Sycamore	*Acer pseudoplatanus* (juv. and seeds)				[85]
2	Common box elder	*Acer negundo* (juv.)				[38]
3	Monkshood	*Aconitum napellus*	[86]			
4	Parsnip	*Carum carvi*	[87]			
5	Sweet grass	*Glyceria*	[87,88]			
6	Cat’s ears	*Hypochaeris radicata*				[89]
7	Parsnip	*Pastinaca sativa*				[36]
8	Bracken	*Pteridium aquilinium*	[90]			
9	Tansy ragwort	*Senecio jacobaea*	[64]			[64]
10	Nettle	*Urtica dioica*			[37]	
		Cases (n)	11			

Number of affected animal individuals (n): beige: n < 10; light red: n > 10; orange: n > 100.

**Table 8 animals-13-03795-t008:** Incidents of poisoning related to co-ingestion of grass, indicated by reference numbers.

No.	Plant Species, Common Name	Plant Species, Scientific Name	Cattle	Sheep	Goats	Horses
1	Sycamore	*Acer pseudoplatanus* (juv.)				[54]
2	Marsh horsetail	*Equisetum palustre*	[87]			
3	Alpine ragwort	*Senecio alpinus*	[91]			
4	Tansy ragwort	*Senecio jacobaea*				[92]
		Cases (n)	4			

Number of affected animal individuals (n): beige: n < 10; light red: n > 10; orange: n > 100.

**Table 9 animals-13-03795-t009:** Incidents of zero poisoning and poisoning under unknown circumstances, indicated by reference numbers.

No.	Plant Species, Common Name	Plant Species, Scientific Name	Cattle	Sheep	Goats	Horses
1	Marsh marigold	*Caltha palustris*				[93]
2	Fat hen	*Chenopodium album*				[94]
3	Cowbane	*Cicuta virosa*	[95]			[96]
4	Autumn crocus	*Colchicum autumnale*	[20]	[19,97]		
5	Horseshoe vetch	*Hippocrepis comosa*	[98]	[98]		
6	Smartweed/knotgrass	*Persicaria/Polygonum*				[26]
7	Bracken	*Pteridium aquilinium*			[28]	
8	Yellow rattle	*Rhinanthus*	[99]			
9	Dock	*Rumex*				[94]
10	Tansy ragwort	*Senecio jacobaea*				[100,101,102]
11	Black nightshade	*Solanum nigrum*			[103]	
12	Yellow oat grass	*Trisetum flavescens*		[104,105]		
13	Nettle	*Urtica dioica*	[106]			
		Cases (n)	16			

Number of affected animal individuals (n): dark green: n = 0; beige: n < 10; orange: n > 100.

## Data Availability

The raw data are deposited at the repository of the Institute of Animal Nutrition of the University of Veterinary Medicine Hannover and will be readily given on demand.

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
