# Peer review of "Do Poisonous Plants in Pastures Communicate Their Toxicity? Meta-Study and Evaluation of Poisoning Cases in Central Europe"

_animals, 2023, doi:10.3390/ani13243795_

Round 1

Reviewer 1 Report

Comments and Suggestions for Authors

Your article caught our full attention, but there were a number of points that caught our eye. A mixture of references of very variable quality, with old and recent data not always fully documented, making metanalysis extremely fragile. The absence of a book or reference that sets the toxic reference (you could have referred to Wagstaff 2008 or Frohne 2006 or BURROWS and TYRL 2013 (even though it deals with toxic plants from North America) would have made it possible to set a framework and rule out plants without reliable, documented data (e.g. Rhinanthus, Polygonum Persicaria...). Data on toxic plants show the importance of vegetative stages, the accessibility of plant organs (roots, stems, fruits, seeds) and the variability of concentrations of toxic principles (Senecio from weakly toxic to highly toxic (Stegelmeier 2011 or Green 2023), which has major consequences on the sensitivity of herbivore species and even different toxicities between ruminants and monogastrics. Older data are interesting for establishing possible toxicity, but only recent data often document cases correctly and attribute them. In addition, stress conditions in certain plants or inhibition by pesticides make weighting and comparison impossible.

On one of your references (very interesting), a toxicological imputation study has been carried out by our services: caraway is indeed imputed for phototoxicity (so spasmolytix activity of stem only seed), but other plants in your list are suspected of containing cyanogenetic heterosides (e.g. Trifolium, Lotus, especially in these spring periods, and the volume unit imputed to Glyceria is missing) and for irritancy Ranunculus could also have been suspected.

Aboling, S.; Rottmann, S.; Wolf, P.; Jahn-Falk, D.; Kamphues1, J. Case Report: Complex Plant Poisoning in Heavily Pregnant 912 Heifers in Germany. Journal of Veterinary Science and Technology 2014, 5, doi:http://dx.doi.org/10.4172/2157-7579.1000178 

references that are difficult to evaluate and which you could perhaps have indexed as a personal communication

20. Aboling, S. Photodermatitis beim Pferd durch Pastinak auf der Weide. Botanische Untersuchung 2011, Unpublished. 824 

21. Aboling, S. Gesichtsschwellung bei einer Ziege durch Äsung von Brennessel. Fallberichts-Sammlung 2022, Unpublished. 

Conclusion

Given the uncertainties and heterogeneity of the data, I think your conclusions need to be more nuanced, because in our science truths are fuzzy and often strewn with exceptions.

Author Response

Dear reviewer,

thank you for your support.

I have carefully checked all comments and suggestions and anwered detailed directly in the PDF file I have created (s. attached).

The citations in the PDF of the revised manuscript shall serve for your convinience, however, they may slightly deviate from the last version of the manuscript because I went again over it and changed here and there some minor issues in the last minute. Please check for the latest changes the revised version I submitted here.

Sincerely yours

The author

Reviewer 2 Report

Comments and Suggestions for Authors

Please see the attached. I think this is an excellent contribution. I realize my suggestions would require a significant reorganizing of the material presented, but I hope the author will consider doing it. I would prefer that the information be presented in its current form rather than not at all, but I also feel it could be of so much greater value to a wider readership, and potentially actually save animal lives, if re-organized and presented first by plant and then by animal type. If my comments are unclear I would be very happy to work directly with the author to better convey what I mean, to help bring out the most from the excellent work conducted.

Comments on the Quality of English Language

A language editor will be able to streamline the manuscript. However, before that, I do think it important for the author to clarify and more accurately convey the different poisoning categories.

Author Response

Dear reviewer,

thank you for your support.

I have carefully checked all comments and suggestions and anwered detailed directly in the PDF file I have created (s. attached). I put two files together - you will see them in one PDF, seperated from each other by distinct pages.

The citations of the revised manuscript in the PDF "author's reply") shall serve for your convinience, however, they may slightly deviate from the last version of the manuscript because I went again over it and changed here and there some minor issues in the last minute. Please check for the latest changes the revised version I submitted here.

Sincerely yours,

The author

Reviewer 3 Report

Comments and Suggestions for Authors

This is an excellent paper with only a few minor corrections required.

Page 3, line 129, Change "poising" to "poisoning"

Page 19, line 729, change "synergetic" to "synergistic"

Page 20, line 751, change "none" to "no"

The paper does not address the possibility that many plant toxins did not evolve to communicate to grazing animals that the plant should not be eaten, but to repel or kill insects, or alternatively to suppress the growth of competing plants. As an example, caffeine is thought to have evolved because it suppresses germination of competing plants and is also toxic to insects that might otherwise eat the beans. I respectfully suggest that the author include a paragraph addressing this possibility, and that repelling grazing by vertebrates is not necessarily the primary reason for the synthesis of toxins by plants.  

Author Response

Dear reviewer,

thank you for your support.

I have carefully checked all comments and suggestions and anwered detailed directly in the PDF file I have created (s. attached).

The citations of the revised manuscript in the PDF "author's reply") shall serve for your convinience, however, they may slightly deviate from the last version of the manuscript because I went again over it and changed here and there some minor issues in the last minute. Please check for the latest changes the revised version I submitted here.

Sincerely yours,

The author

Round 2

Reviewer 2 Report

Comments and Suggestions for Authors

I think this is a very valuable contribution! The addition of the common names and the supplementary materials is very much appreciated as it makes the information just that much more accessible to readers.

Comments on the Quality of English Language

French is my second language so to me this revised version reads fairly well, though it does still need some revision by an editor to smooth out some of the rough edges around grammar.

Author Response

Dear reviewer,

again, I am grateful for your excellent support and your extensive improvement of the manuscript.

Thank you for your time and great input and the exchange of thoughts!

Having done this metastudy alone (it is an invited review), I have much appreciated the stimulating discsussion on the topic with you. I will miss this kind of expertise, your keen interest and the questions you arised.

Please find my comments to your second review in the PDF file.

Kind regards from

the author

Round 3

Reviewer 2 Report

Comments and Suggestions for Authors

Congratulations on this paper, it is a very important contribution and the information you gathered can directly protect livestock from plant poisoning. The focus on plant-animal communication is really interesting, and I feel that deeper angle (animals having to ignore a message of toxicity due to hunger) might compel veterinarians and farmers in a unique way to offer sufficient, safe feed in pastures.

Two final comments, but please take with a grain of salt:

Abstract:

at the end of the first sentence you could add the words '...in order to secure co-existence'.

However I understand word count might be limited and you may feel this is repetitive. To me this would just clarify one of your main points, which is that co-existence can only work if a) animals can perceive the message of toxicity and b) heed it by not ingesting the plant.

As a very last sentence, you could conclude by saying:

The checklist developed during this review provides a tool which veterinarians and farmers may consult to ensure that animals grazing in pastures have sufficient appropriate feed choices to heed the communication by resident poisonous plants and avoid eating them.

I look forward to seeing the paper in published form!

Comments on the Quality of English Language

The paper would just need a final scan-through by the journal's language editor to standardize the English and clarify a few grammatical inconsistencies. Otherwise it reads well.